# HiFi-GAN: Generative Adversarial Networks for Efficient and High Fidelity Speech Synthesis

**Jungil Kong**
Kakao Enterprise
henry.k@kakaoenterprise.com

**Jaehyeon Kim**
Kakao Enterprise
jay.xyz@kakaoenterprise.com

**Jaekyoung Bae**
Kakao Enterprise
storm.b@kakaoenterprise.com

## Abstract

Several recent work on speech synthesis have employed generative adversarial networks (GANs) to produce raw waveforms. Although such methods improve the sampling efficiency and memory usage, their sample quality has not yet reached that of autoregressive and flow-based generative models. In this work, we propose HiFi-GAN, which achieves both efficient and high-fidelity speech synthesis. As speech audio consists of sinusoidal signals with various periods, we demonstrate that modeling periodic patterns of an audio is crucial for enhancing sample quality. A subjective human evaluation (mean opinion score, MOS) of a single speaker dataset indicates that our proposed method demonstrates similarity to human quality while generating 22.05 kHz high-fidelity audio 167.9 times faster than real-time on a single V100 GPU. We further show the generality of HiFi-GAN to the mel-spectrogram inversion of unseen speakers and end-to-end speech synthesis. Finally, a small footprint version of HiFi-GAN generates samples 13.4 times faster than real-time on CPU with comparable quality to an autoregressive counterpart.

## 1   Introduction

Voice is one of the most frequent and naturally used communication interfaces for humans. With recent developments in technology, voice is being used as a main interface in artificial intelligence (AI) voice assistant services such as Amazon Alexa, and it is also widely used in automobiles, smart homes and so forth. Accordingly, with the increase in demand for people to converse with machines, technology that synthesizes natural speech like human speech is being actively studied.

Recently, with the development of neural networks, speech synthesis technology has made a rapid progress. Most neural speech synthesis models use a two-stage pipeline: 1) predicting a low resolution intermediate representation such as mel-spectrograms (Shen et al., 2018, Ping et al., 2017, Li et al., 2019) or linguistic features (Oord et al., 2016) from text, and 2) synthesizing raw waveform audio from the intermediate representation (Oord et al., 2016, 2018, Prenger et al., 2019, Kumar et al., 2019). The first stage is to model low-level representations of human speech from text, whereas the second stage model synthesizes raw waveforms with up to 24,000 samples per second and up to 16 bit fidelity. In this work, we focus on designing a second stage model that efficiently synthesizes high fidelity waveforms from mel-spectrograms.

Various work have been conducted to improve the audio synthesis quality and efficiency of second stage models. WaveNet (Oord et al., 2016) is an autoregressive (AR) convolutional neural network that demonstrates the ability of neural network based methods to surpass conventional methods in quality.

However, because of the AR structure, WaveNet generates one sample at each forward operation; it is prohibitively slow in synthesizing high temporal resolution audio. Flow-based generative models are proposed to address this problem. Because of their ability to model raw waveforms by transforming noise sequences of the same size in parallel, flow-based generative models fully utilize modern parallel computing processors to speed up sampling. Parallel WaveNet (Oord et al., 2018) is an inverse autoregressive flow (IAF) that is trained to minimize its Kullback-Leibler divergence from a pre-trained WaveNet called a teacher. Compared to the teacher model, it improves the synthesis speed to 1,000 times or more, without quality degradation. WaveGlow (Prenger et al., 2019) eliminates the need for distilling a teacher model, and simplifies the learning process through maximum likelihood estimation by employing efficient bijective flows based on Glow (Kingma and Dhariwal, 2018). It also produces high-quality audio compared to WaveNet. However, it requires many parameters for its deep architecture with over 90 layers.

Generative adversarial networks (GANs) (Goodfellow et al., 2014), which are one of the most dominant deep generative models, have also been applied to speech synthesis. Kumar et al. (2019) proposed a multi-scale architecture for discriminators operating on multiple scales of raw waveforms. With sophisticated architectural consideration, the MelGAN generator is compact enough to enable real-time synthesis on CPU. Yamamoto et al. (2020) proposed multi-resolution STFT loss function to improve and stabilize GAN training and achieved better parameter efficiency and less training time than an IAF model, ClariNet (Ping et al., 2018). Instead of mel-spectrograms, GAN-TTS (Bińkowski et al., 2019) successfully generates high quality raw audio waveforms from linguistic features through multiple discriminators operating on different window sizes. The model also shows fewer FLOPs compared to Parallel WaveNet. Despite the advantages, there is still a gap in sample quality between the GAN models and AR or flow-based models.

We propose HiFi-GAN, which achieves both higher computational efficiency and sample quality than AR or flow-based models. As speech audio consists of sinusoidal signals with various periods, modeling the periodic patterns matters to generate realistic speech audio. Therefore, we propose a discriminator which consists of small sub-discriminators, each of which obtains only a specific periodic parts of raw waveforms. This architecture is the very ground of our model successfully synthesizing realistic speech audio. As we extract different parts of audio for the discriminator, we also design a module that places multiple residual blocks each of which observes patterns of various lengths in parallel, and apply it to the generator.

HiFi-GAN achieves a higher MOS score than the best publicly available models, WaveNet and WaveGlow. It synthesizes human-quality speech audio at speed of 3.7 MHz on a single V100 GPU. We further show the generality of HiFi-GAN to the mel-spectrogram inversion of unseen speakers and end-to-end speech synthesis. Finally, the tiny footprint version of HiFi-GAN requires only 0.92M parameters while outperforming the best publicly available models and the fastest version of HiFi-GAN samples 13.44 times faster than real-time on CPU and 1,186 times faster than real-time on single V100 GPU with comparable quality to an autoregressive counterpart.

Our audio samples are available on the demo web-site[1], and we provide the implementation as open source for reproducibility and future work.[2]

## 2 HiFi-GAN

### 2.1 Overview

HiFi-GAN consists of one generator and two discriminators: multi-scale and multi-period discriminators. The generator and discriminators are trained adversarially, along with two additional losses for improving training stability and model performance.

### 2.2 Generator

The generator is a fully convolutional neural network. It uses a mel-spectrogram as input and upsamples it through transposed convolutions until the length of the output sequence matches the temporal resolution of raw waveforms. Every transposed convolution is followed by a multi-receptive

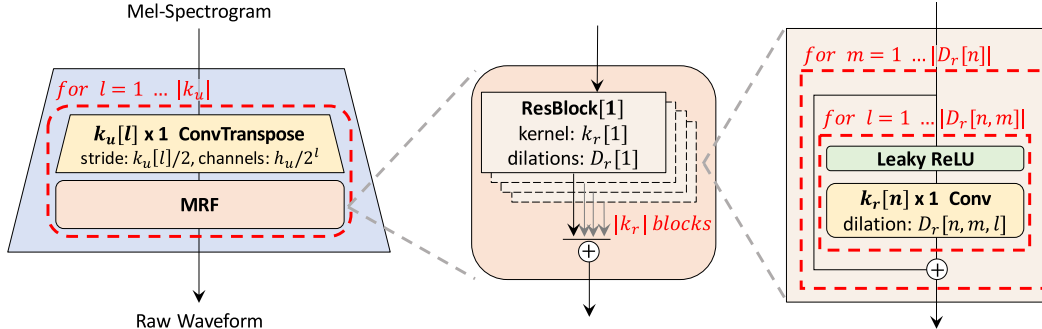

Figure 1: The generator upsamples mel-spectrograms up to $|k_u|$ times to match the temporal resolution of raw waveforms. A MRF module adds features from $|k_r|$ residual blocks of different kernel sizes and dilation rates. Lastly, the $n$-th residual block with kernel size $k_r[n]$ and dilation rates $D_r[n]$ in a MRF module is depicted.

field fusion (MRF) module, which we describe in the next paragraph. Figure 1 shows the architecture of the generator. As in previous work (Mathieu et al., 2015, Isola et al., 2017, Kumar et al., 2019), noise is not given to the generator as an additional input.

**Multi-Receptive Field Fusion** We design the multi-receptive field fusion (MRF) module for our generator, which observes patterns of various lengths in parallel. Specifically, MRF module returns the sum of outputs from multiple residual blocks. Different kernel sizes and dilation rates are selected for each residual block to form diverse receptive field patterns. The architectures of MRF module and a residual block are shown in Figure 1. We left some adjustable parameters in the generator; the hidden dimension $h_u$, kernel sizes $k_u$ of the transposed convolutions, kernel sizes $k_r$, and dilation rates $D_r$ of MRF modules can be regulated to match one's own requirement in a trade-off between synthesis efficiency and sample quality.

## 2.3 Discriminator

Identifying long-term dependencies is the key for modeling realistic speech audio. For example, a phoneme duration can be longer than 100 ms, resulting in high correlation between more than 2,200 adjacent samples in the raw waveform. This problem has been addressed in the previous work (Donahue et al., 2018) by increasing receptive fields of the generator and discriminator. We focus on another crucial problem that has yet been resolved; as speech audio consists of sinusoidal signals with various periods, the diverse periodic patterns underlying in the audio data need to be identified.

To this end, we propose the multi-period discriminator (MPD) consisting of several sub-discriminators each handling a portion of periodic signals of input audio. Additionally, to capture consecutive patterns and long-term dependencies, we use the multi-scale discriminator (MSD) proposed in MelGAN (Kumar et al., 2019), which consecutively evaluates audio samples at different levels. We conducted simple experiments to show the ability of MPD and MSD to capture periodic patterns, and the results can be found in Appendix B.

**Multi-Period Discriminator** MPD is a mixture of sub-discriminators, each of which only accepts equally spaced samples of an input audio; the space is given as period $p$. The sub-discriminators are designed to capture different implicit structures from each other by looking at different parts of an input audio. We set the periods to [2, 3, 5, 7, 11] to avoid overlaps as much as possible. As shown in Figure 2b, we first reshape 1D raw audio of length $T$ into 2D data of height $T/p$ and width $p$ and then apply 2D convolutions to the reshaped data. In every convolutional layer of MPD, we restrict the kernel size in the width axis to be 1 to process the periodic samples independently. Each sub-discriminator is a stack of strided convolutional layers with leaky rectified linear unit (ReLU) activation. Subsequently, weight normalization (Salimans and Kingma, 2016) is applied to MPD. By reshaping the input audio into 2D data instead of sampling periodic signals of audio, gradients from MPD can be delivered to all time steps of the input audio.

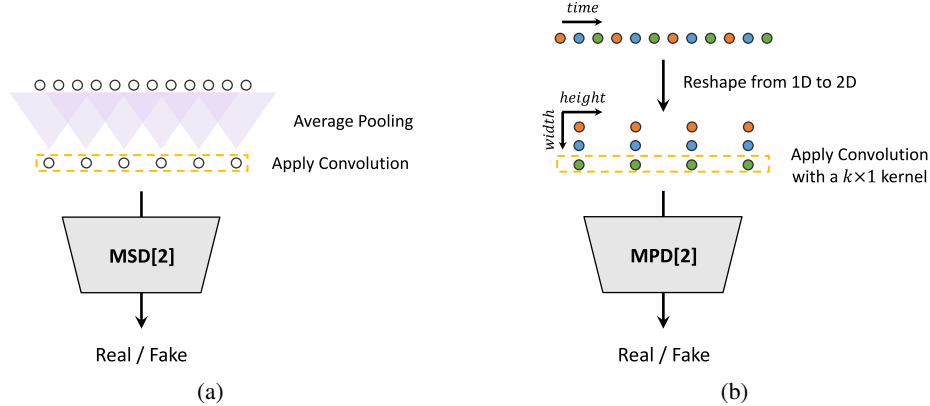

Figure 2: (a) The second sub-discriminator of MSD. (b) The second sub-discriminator of MPD with period 3.

**Multi-Scale Discriminator**   Because each sub-discriminator in MPD only accepts disjoint samples, we add MSD to consecutively evaluate the audio sequence. The architecture of MSD is drawn from that of MelGAN (Kumar et al., 2019). MSD is a mixture of three sub-discriminators operating on different input scales: raw audio, ×2 average-pooled audio, and ×4 average-pooled audio, as shown in Figure 2a. Each of the sub-discriminators in MSD is a stack of strided and grouped convolutional layers with leaky ReLU activation. The discriminator size is increased by reducing stride and adding more layers. Weight normalization is applied except for the first sub-discriminator, which operates on raw audio. Instead, spectral normalization (Miyato et al., 2018) is applied and stabilizes training as it reported.

Note that MPD operates on disjoint samples of raw waveforms, whereas MSD operates on smoothed waveforms.

For previous work using multi-discriminator architecture such as MPD and MSD, Bińkowski et al. (2019)'s work can also be referred. The discriminator architecture proposed in the work has resemblance to MPD and MSD in that it is a mixture of discriminators, but MPD and MSD are Markovian window-based fully unconditional discriminator, whereas it averages the output and has conditional discriminators. Also, the resemblance between MPD and RWD (Bińkowski et al., 2019) can be considered in the part of reshaping input audio, but MPD uses periods set to prime numbers to discriminate data of as many periods as possible, whereas RWD uses reshape factors of overlapped periods and does not handle each channel of the reshaped data separately, which are different from what MPD is aimed for. A variation of RWD can perform a similar operation to MPD, but it is also not the same as MPD in terms of parameter sharing and strided convolution to adjacent signals. More details of the architectural difference can be found in Appendix C.

## 2.4   Training Loss Terms

**GAN Loss**   For brevity, we describe our discriminators, MSD and MPD, as one discriminator throughout Section 2.4. For the generator and discriminator, the training objectives follow LS-GAN (Mao et al., 2017), which replace the binary cross-entropy terms of the original GAN objectives (Goodfellow et al., 2014) with least squares loss functions for non-vanishing gradient flows. The discriminator is trained to classify ground truth samples to 1, and the samples synthesized from the generator to 0. The generator is trained to fake the discriminator by updating the sample quality to be classified to a value almost equal to 1. GAN losses for the generator $G$ and the discriminator $D$ are defined as

$$\mathcal{L}_{Adv}(D;G) = \mathbb{E}_{(x,s)}\left[(D(x)-1)^2 + (D(G(s)))^2\right] \tag{1}$$

$$\mathcal{L}_{Adv}(G;D) = \mathbb{E}_s\left[(D(G(s))-1)^2\right] \tag{2}$$

, where $x$ denotes the ground truth audio and $s$ denotes the input condition, the mel-spectrogram of the ground truth audio.

**Mel-Spectrogram Loss**   In addition to the GAN objective, we add a mel-spectrogram loss to improve the training efficiency of the generator and the fidelity of the generated audio. Referring to previous work (Isola et al., 2017), applying a reconstruction loss to GAN model helps to generate realistic results, and in Yamamoto et al. (2020)'s work, time-frequency distribution is effectively captured by jointly optimizing multi-resolution spectrogram and adversarial loss functions. We used mel-spectrogram loss according to the input conditions, which can also be expected to have the effect of focusing more on improving the perceptual quality due to the characteristics of the human auditory system. The mel-spectrogram loss is the L1 distance between the mel-spectrogram of a waveform synthesized by the generator and that of a ground truth waveform. It is defined as

$$\mathcal{L}_{Mel}(G) = \mathbb{E}_{(x,s)}\left[||\phi(x) - \phi(G(s))||_1\right] \tag{3}$$

, where $\phi$ is the function that transforms a waveform into the corresponding mel-spectrogram. The mel-spectrogram loss helps the generator to synthesize a realistic waveform corresponding to an input condition, and also stabilizes the adversarial training process from the early stages.

**Feature Matching Loss**   The feature matching loss is a learned similarity metric measured by the difference in features of the discriminator between a ground truth sample and a generated sample (Larsen et al., 2016, Kumar et al., 2019). As it was successfully adopted to speech synthesis (Kumar et al., 2019), we use it as an additional loss to train the generator. Every intermediate feature of the discriminator is extracted, and the L1 distance between a ground truth sample and a conditionally generated sample in each feature space is calculated. The feature matching loss is defined as

$$\mathcal{L}_{FM}(G; D) = \mathbb{E}_{(x,s)}\left[\sum_{i=1}^{T}\frac{1}{N_i}||D^i(x) - D^i(G(s))||_1\right] \tag{4}$$

, where $T$ denotes the number of layers in the discriminator; $D^i$ and $N_i$ denote the features and the number of features in the $i$-th layer of the discriminator, respectively.

**Final Loss**   Our final objectives for the generator and discriminator are as

$$\mathcal{L}_G = \mathcal{L}_{Adv}(G; D) + \lambda_{fm}\mathcal{L}_{FM}(G; D) + \lambda_{mel}\mathcal{L}_{Mel}(G) \tag{5}$$

$$\mathcal{L}_D = \mathcal{L}_{Adv}(D; G) \tag{6}$$

, where we set $\lambda_{fm} = 2$ and $\lambda_{mel} = 45$. Because our discriminator is a set of sub-discriminators of MPD and MSD, Equations 5 and 6 can be converted with respect to the sub-discriminators as

$$\mathcal{L}_G = \sum_{k=1}^{K}\left[\mathcal{L}_{Adv}(G; D_k) + \lambda_{fm}\mathcal{L}_{FM}(G; D_k)\right] + \lambda_{mel}\mathcal{L}_{Mel}(G) \tag{7}$$

$$\mathcal{L}_D = \sum_{k=1}^{K}\mathcal{L}_{Adv}(D_k; G) \tag{8}$$

, where $D_k$ denotes the $k$-th sub-discriminator in MPD and MSD.

## 3   Experiments

For fair and reproducible comparison with other models, we used the LJSpeech dataset (Ito, 2017) in which many speech synthesis models are trained. LJSpeech consists of 13,100 short audio clips of a single speaker with a total length of approximately 24 hours. The audio format is 16-bit PCM with a sample rate of 22 kHz; it was used without any manipulation. HiFi-GAN was compared against the best publicly available models: the popular mixture of logistics (MoL) WaveNet (Oord et al., 2018)

implementation (Yamamoto, 2018) [3] and the official implementation of WaveGlow (Valle, 2018b) and MelGAN (Kumar, 2019). We used the provided pretrained weights for all the models.

To evaluate the generality of HiFi-GAN to the mel-spectrogram inversion of unseen speakers, we used the VCTK multi-speaker dataset (Veaux et al., 2017), which consists of approximately 44,200 short audio clips uttered by 109 native English speakers with various accents. The total length of the audio clips is approximately 44 hours. The audio format is 16-bit PCM with a sample rate of 44 kHz. We reduced the sample rate to 22 kHz. We randomly selected nine speakers and excluded all their audio clips from the training set. We then trained MoL WaveNet, WaveGlow, and MelGAN with the same data settings; all the models were trained until 2.5M steps.

To evaluate the audio quality, we crowd-sourced 5-scale MOS tests via Amazon Mechanical Turk. The MOS scores were recorded with 95% confidence intervals (CI). Raters listened to the test samples randomly, where they were allowed to evaluate each audio sample once. All audio clips were normalized to prevent the influence of audio volume differences on the raters. All quality assessments in Section 4 were conducted in this manner, and were not sourced from other papers.

The synthesis speed was measured on GPU and CPU environments according to the recent research trends regarding the efficiency of neural networks (Kumar et al., 2019, Zhai et al., 2020, Tan et al., 2020). The devices used are a single NVIDIA V100 GPU and a MacBook Pro laptop (Intel i7 CPU 2.6GHz). Additionally, we used 32-bit floating point operations for all the models without any optimization methods.

To confirm the trade-off between synthesis efficiency and sample quality, we conducted experiments based on the three variations of the generator, $V1$, $V2$, and $V3$ while maintaining the same discriminator configuration. For $V1$, we set $h_u = 512$, $k_u = [16, 16, 4, 4]$, $k_r = [3, 7, 11]$, and $D_r = [[1, 1], [3, 1], [5, 1]] \times 3]$. $V2$ is simply a smaller version of $V1$, which has a smaller hidden dimension $h_u = 128$ but with exactly the same receptive fields. To further reduce the number of layers while maintaining receptive fields wide, the kernel sizes and dilation rates of $V3$ were selected carefully. The detailed configurations of the models are listed in Appendix A.1. We used 80 bands mel-spectrograms as input conditions. The FFT, window, and hop size were set to 1024, 1024, and 256, respectively. The networks were trained using the AdamW optimizer (Loshchilov and Hutter, 2017) with $\beta_1 = 0.8$, $\beta_2 = 0.99$, and weight decay $\lambda = 0.01$. The learning rate decay was scheduled by a 0.999 factor in every epoch with an initial learning rate of $2 \times 10^{-4}$.

## 4   Results

### 4.1   Audio Quality and Synthesis Speed

To evaluate the performance of our models in terms of both quality and speed, we performed the MOS test for spectrogram inversion, and the speed measurement. For the MOS test, we randomly selected 50 utterances from the LJSpeech dataset and used the ground truth spectrograms of the utterances which were excluded from training as input conditions.

For easy comparison of audio quality, synthesis speed and model size, the results are compiled and presented in Table1. Remarkably, all variations of HiFi-GAN scored higher than the other models. $V1$ has 13.92M parameters and achieves the highest MOS with a gap of 0.09 compared to the ground truth audio; this implies that the synthesized audio is nearly indistinguishable from the human voice. In terms of synthesis speed, $V1$ is faster than WaveGlow and MoL WaveNet. $V2$ also demonstrates similarity to human quality with a MOS of 4.23 while significantly reducing the memory requirement and inference time, compared to $V1$. It only requires 0.92M parameters. Despite having the lowest MOS among our models, $V3$ can synthesize speech 13.44 times faster than real-time on CPU and 1,186 times faster than real-time on single V100 GPU while showing similar perceptual quality with MoL WaveNet. As $V3$ efficiently synthesizes speech on CPU, it can be well suited for on-device applications.

Table 1: Comparison of the MOS and the synthesis speed. Speed of $n$ kHz means that the model can generate $n \times 1000$ raw audio samples per second. The numbers in () mean the speed compared to real-time.

| Model | MOS (CI) | Speed on CPU (kHz) | | Speed on GPU (kHz) | | # Param (M) |
|---|---|---|---|---|---|---|
| Ground Truth | 4.45 ($\pm$0.06) | $-$ | | $-$ | | $-$ |
| WaveNet (MoL) | 4.02 ($\pm$0.08) | $-$ | | 0.07 | ($\times$0.003) | 24.73 |
| WaveGlow | 3.81 ($\pm$0.08) | 4.72 | ($\times$0.21) | 501 | ($\times$22.75) | 87.73 |
| MelGAN | 3.79 ($\pm$0.09) | 145.52 | ($\times$6.59) | 14,238 | ($\times$645.73) | 4.26 |
| HiFi-GAN $V1$ | **4.36** ($\pm$0.07) | 31.74 | ($\times$1.43) | 3,701 | ($\times$167.86) | 13.92 |
| HiFi-GAN $V2$ | 4.23 ($\pm$0.07) | 214.97 | ($\times$9.74) | 16,863 | ($\times$764.80) | **0.92** |
| HiFi-GAN $V3$ | 4.05 ($\pm$0.08) | **296.38** | **($\times$13.44)** | **26,169** | **($\times$1,186.80)** | 1.46 |

## 4.2 Ablation Study

We performed an ablation study of MPD, MRF, and mel-spectrogram loss to verify the effect of each HiFi-GAN component on the quality of the synthesized audio. $V3$ that has the smallest expressive power among the three generator variations was used as a generator for the ablation study, and the network parameters were updated up to 500k steps for each configuration.

The results of the MOS evaluation are shown in Table 2, which show all three components contribute to the performance. Removing **MPD** causes a significant decrease in perceptual quality, whereas the absence of **MSD** shows a relatively small but noticeable degradation. To investigate the effect of **MRF**, one residual block with the widest receptive field was retained in each MRF module. The result is also worse than the baseline. The experiment on the **mel-spectrogram loss** shows that it helps improve the quality, and we observed that the quality improves more stably when the loss is applied.

To verify the effect of MPD in the settings of other GAN models, we introduced MPD in MelGAN. MelGAN trained with MPD outperforms the original one by a gap of 0.47 MOS, which shows statistically significant improvement.

We experimented with periods of powers of 2 to verify the effect of periods set to prime numbers. While the period 2 allows signals to be processed closely, it results in statistically significant degradation with a difference of 0.20 MOS from the baseline.

Table 2: Ablation study results. Comparison of the effect of each component on the synthesis quality.

| Model | MOS (CI) |
|---|---|
| Ground Truth | 4.57 ($\pm$0.04) |
| Baseline (HiFi-GAN $V3$) | 4.10 ($\pm$0.05) |
| w/o MPD | 2.28 ($\pm$0.09) |
| w/o MSD | 3.74 ($\pm$0.05) |
| w/o MRF | 3.92 ($\pm$0.05) |
| w/o Mel-Spectrogram Loss | 3.25 ($\pm$0.05) |
| MPD $p$=[2,4,8,16,32] | 3.90 ($\pm$0.05) |
| MelGAN | 2.88 ($\pm$0.08) |
| MelGAN with MPD | 3.35 ($\pm$0.07) |

## 4.3 Generalization to Unseen Speakers

We used 50 randomly selected utterances of nine unseen speakers in the VCTK dataset that were excluded from the training set for the MOS test. Table 3 shows the experimental results for the mel-spectrogram inversion of the unseen speakers. The three generator variations scored 3.77, 3.69, and 3.61. They were all better than AR and flow-based models, indicating that the proposed models

generalize well to unseen speakers. Additionally, the tendency of difference in MOS scores of the proposed models is similar with the result shown in Section 4.1, which exhibits generalization across different datasets.

Table 3: Quality comparison of synthe-sized utterances for unseen speakers.

| Model | MOS (CI) |
|---|---|
| Ground Truth | 3.79 ($\pm$0.07) |
| WaveNet (MoL) | 3.52 ($\pm$0.08) |
| WaveGlow | 3.52 ($\pm$0.08) |
| MelGAN | 3.50 ($\pm$0.08) |
| HiFi-GAN $V1$ | **3.77** ($\pm$0.07) |
| HiFi-GAN $V2$ | 3.69 ($\pm$0.07) |
| HiFi-GAN $V3$ | 3.61 ($\pm$0.07) |

Table 4: Quality comparison for end-to-end speech synthesis.

| Model | MOS (CI) |
|---|---|
| Ground Truth | 4.23 ($\pm$0.07) |
| WaveGlow (w/o fine-tuning) | 3.69 ($\pm$0.08) |
| HiFi-GAN $V1$ (w/o fine-tuning) | 3.91 ($\pm$0.08) |
| HiFi-GAN $V2$ (w/o fine-tuning) | 3.88 ($\pm$0.08) |
| HiFi-GAN $V3$ (w/o fine-tuning) | 3.89 ($\pm$0.08) |
| WaveGlow (find-tuned) | 3.66 ($\pm$0.08) |
| HiFi-GAN $V1$ (find-tuned) | **4.18** ($\pm$0.08) |
| HiFi-GAN $V2$ (find-tuned) | 4.12 ($\pm$0.07) |
| HiFi-GAN $V3$ (find-tuned) | 4.02 ($\pm$0.08) |

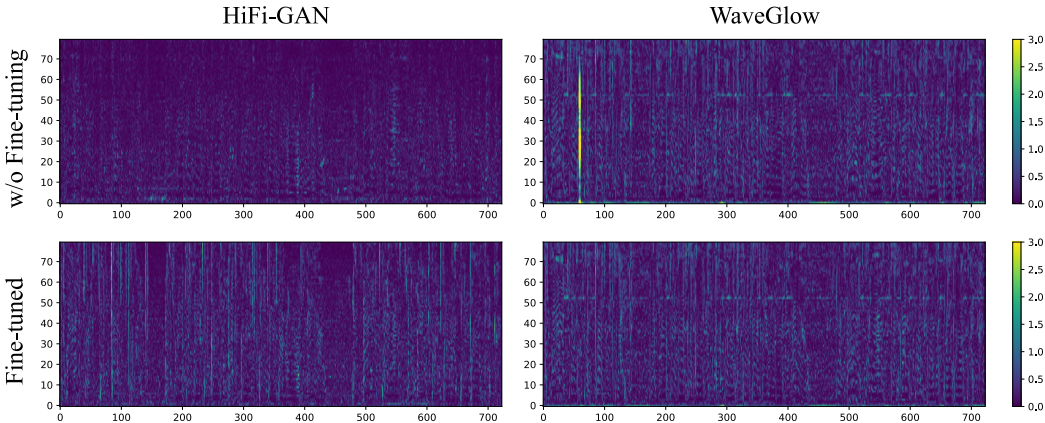

Figure 3: Pixel-wise difference in the mel-spectrogram domain between generated waveforms and a mel-spectrogram from Tacotron2. Before fine-tuning, HiFi-GAN generates waveforms corresponding to input conditions accurately. After fine-tuning, the error of the mel-spectrogram level increased, but the perceptual quality increased.

## 4.4 End-to-End Speech Synthesis

We conducted an additional experiment to examine the effectiveness of the proposed models when applied to an end-to-end speech synthesis pipeline, which consists of *text to mel-spectrogram* and *mel-spectrogram to waveform* synthesis modules. We herein used Tacotron2 (Shen et al., 2018) to generate mel-spectrograms from text. Without any modification, we synthesized the mel-spectrograms using the most popular implementation of Tacotron2 (Valle, 2018a) with the provided pre-trained weights. We then fed them as input conditions into second stage models, including our models and WaveGlow used in Section 4.1.[4]

The MOS scores are listed in Table 4. The results without fine-tuning show that all the proposed models outperform WaveGlow in the end-to-end setting, while the audio quality of all models are unsatisfactory compared to the ground truth audio. However, when the pixel-wise difference in the mel-spectrogram domain between generated waveforms and a mel-spectrogram from Tacotron2 are

investigated as demonstrated in Figure 3, we found that the difference is insignificant, which means that the predicted mel-spectrogram from Tacotron2 was already noisy. To improve the audio quality in the end-to-end setting, we applied fine-tuning with predicted mel-spectrograms of Tacotron2 in teacher-forcing mode (Shen et al., 2018) to all the models up to 100k steps. MOS scores of all the fine-tuned proposed models over 4, whereas fine-tuned WaveGlow did not show quality improvement. We conclude that HiFi-GAN adapts well on the end-to-end setting with fine-tuning.

## 5   Conclusion

In this work, we introduced HiFi-GAN, which can efficiently synthesize high quality speech audio. Above all, our proposed model outperforms the best performing publicly available models in terms of synthesis quality, even comparable to human level. Moreover, it shows a significant improvement in terms of synthesis speed. We took inspiration from the characteristic of speech audio that consists of patterns with various periods and applied it to neural networks, and verified that the existence of the proposed discriminator greatly influences the quality of speech synthesis through the ablation study. Additionally, this work presents several experiments that are significant in speech synthesis applications. HiFi-GAN shows excellent ability to generalize unseen speakers and synthesize speech audio comparable to human quality from noisy inputs in an end-to-end setting. In addition, our small footprint model demonstrates comparable sample quality with the best publicly available autoregressive counterpart, while generating samples in an order-of-magnitude faster than real-time on CPU. This shows progress towards on-device natural speech synthesis, which requires low latency and memory footprint. Finally, our experiments show that the generators of various configurations can be trained with the same discriminators and learning mechanism, which indicates the possibility of flexibly selecting a generator configuration according to the target specifications without the need for a time-consuming hyper-parameter search for the discriminators.

We release HiFi-GAN as open source. We envisage that our work will serve as a basis for future speech synthesis studies.

## Broader Impact

Speech synthesis technology has been developed for a long time, and recently, it has reached the level of producing speech at human level by applying neural networks. However, although the quality of synthesized audio is very high, there are several limitations for applying the neural network based speech synthesis technology in real production due to the high computation cost and slow synthesis speed. Speech synthesis models that perform high speed synthesis have also been studied, but the quality of these models needed to be improved compared to that of human.

Our work addressed these problems, and showed that the proposed model achieves both high quality and fast speech synthesis. This could bring advantages to service providers who provide voice interfaces. Service providers can improve user satisfaction by providing natural and higher quality speech audio allows listeners to understand content more quickly and accurately. In addition, the high computational efficiency obtained in our work could reduce the usage of computing devices such as GPUs and CPUs, which can contribute to the cost savings of the service providers. It could also enable on-device operation of neural network based speech synthesis, which could lower the service latency and computation cost on servers.

Meanwhile, the high efficiency and quality of the speech samples that our work is capable of generating can accompany certain adverse effects. First, the demand for recordings of professional voice actors may be reduced. In many applications such as subway announcements, recordings of professional voice actors are embedded. So, when a new announcement is needed, a professional voice actor will record the utterance for the announcement. However, if the neural network based speech synthesis model is capable of producing natural speech like humans, it can replace the work of professional voice actors. This could be considered an inevitable change in the job due to technological advancements. However, we need to devise a way such as contracts for the use of voice, not the cost of recording, to mitigate the sudden impact on our society.

Furthermore, high-performance speech synthesis technology can be used for criminal activities such as voice phishing. If a neural network based speech synthesis model was trained with a dataset consisting of speaker characteristics that give someone confidence, such as those of news anchors, it might be used for making people feel secure in the deceptions. In addition, our work has shown that it is possible to generate high-quality speech audio with extremely fast inference speed, which could be used for voice phishing attackers to generate response in real-time during conversation with targets. By combining the speaker characteristics and fast speech synthesis, it might result in increasing the success rate of the voice phishing attacks.

The unintended abuse of voice in datasets could also be a problem. Since the speech synthesis models based on neural networks are trained with audio clips of real human speech, the synthesized audio is also very similar to the voice of a speaker who recorded audio clips in the training data. It would not be a problem if the trained model is only to be used in a controlled environment. However, if the model is stolen by a malicious person or if the speech synthesis model is released as a service to allow synthesizing any utterances without restrictions, it could be abused regardless of the original intention of the voice owners or the model owner. Therefore, we believe that institutes and companies that study and use speech synthesis models should pay particular attention to the security of the models and the training datasets.

The results of our model failure could be audio clips with noise, distortion, or inaccurate or missing pronunciation. These issues might be acceptable in most cases, but could be a problem when the model is applied to critical tasks such as emergencies or applications designed to listen to audio only once. As is known, most neural network based models are not fully controllable. Therefore, it is difficult to guarantee that it works 100% robust under any conditions. We believe that neural network based speech synthesis technology should be carefully applied in mission critical domains.

## Acknowledgments and Disclosure of Funding

We would like to thank Bokyung Son, Sungwon Kim, Yongjin Cho and Sungwon Lyu.

All resources used for this work were provided by Kakao Enterprise.

## Footnotes

[1] https://jik876.github.io/hifi-gan-demo/

[2] https://github.com/jik876/hifi-gan

[3]In the original paper (Oord et al., 2018), an input condition called linguistic features was used, but this implementation uses mel-spectrogram as the input condition as in the later paper (Shen et al., 2018).

[4]MelGAN and MoL WaveNet were excluded from comparison for their differences in pre-processing such as frequency clipping. Mismatch in the input representation led to producing low quality audio when combined with Tacotron2.

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
