[Supplementary Material]

# Supplementary Material of
# HiFi-GAN: Generative Adversarial Networks for Efficient and High Fidelity Speech Synthesis

## Appendix A

### A.1. Details of the Model Architecture

The detailed architecture of the generator and MPD is depicted in Figure 4. The configuration of three variants of the generator is listed in Table 5.

Figure 4: (a) The generator. (b) The sub-discriminator of MPD with period $p$.

Table 5: Hyper-parameters of three generator $V1$, $V2$, and $V3$.

| Model | Hyper-Parameters | | | |
|---|---|---|---|---|
| | $h_u$ | $k_u$ | $k_r$ | $D_r$ |
| $V1$ | 512 | [16, 16, 4, 4] | [3, 7, 11] | [[1, 1], [3, 1], [5, 1]] $\times 3$ |
| $V2$ | 128 | [16, 16, 4, 4] | [3, 7, 11] | [[1, 1], [3, 1], [5, 1]] $\times 3$ |
| $V3$ | 256 | [16, 16, 8] | [3, 5, 7] | [[1], [2]], [[2], [6]], [[3], [12]] |

Each ResBlock in the generators is a structure in which multiple convolution layers and residual connections are stacked. In the ResBlock of V1 and V2, 2 convolution layers and 1 residual connection are stacked 3 times. In the Resblock of V3, 1 convolution layer and 1 residual connection are stacked 2 times. Therefore, V3 consists of a much smaller number of layers than V1 and V2.

# Appendix B

## B.1. Periodic signal discrimination experiments

We conducted additional experiments similar to training a discriminator using a simple dataset to verify the ability of MPD to discriminate periodic signals. We have assumed that most of the signals generated by the generator are well made, with a very small amount of error signals. We generated 40,000 sinusoidal data of randomly selected frequency in the range of 1~8,000Hz, random phase and random energy. We experimented with 3 ratios to see the performance change according to the difference between the ratio of true and false label. We gave true label [99, 99.5, 99.9]% of the frequencies and false label the remaining frequencies. We added a projection layer to MPD and MSD, trained with these datasets, and compared the classification results of 8,000 unseen data. As this is binary classification, we set the ratio of true label to false label of validation data to 50:50 to facilitate comparison. We repeated this experiment 5 times to get the average, and the results are listed in Table 6. The results show that MPD is superior in discriminating periodic signals than MSD. And remarkably MPD can discriminate 0.1% of false frequencies with 85% accuracy.

Table 6: Comparison of classification accuracy for periodic signals.

| Model | True Label Ratio | | |
| | 99.0% | 99.5% | 99.9% |
| --- | --- | --- | --- |
| MSD | 80.76% | 59.01% | 50.42% |
| MPD | 97.01% | 96.38% | 85.33% |

## B.2. Frequency response analysis of input signals of discriminators

Figure 5: (a) The ground-truth sinc function. (b) The down-sampled input signals for sub-discriminators and their frequency responses. (c) The synthesized signals. (b) The down-sampled synthesized signals for sub-discriminators and their frequency responses.

We investigated the importance of modeling discriminators to capture periodic patterns through a toy example. In this example, we train two generators to mimic the sinc function each with MPD and MSD, respectively. we set the domain of sinc function to evenly spaced 1,000 numbers over an interval [-200, 200], and set the function values as the ground-truth signal, which is visualized in Figure 5a. To maintain the equal structure of MSD and MPD, we set the periods of MPD to [1, 2, 4]. All sub-discriminators of MPD and MSD are designed as feed-forward networks with the same structure. The generator is simply modeled with 1,000 parameters each corresponded to one point of the domain of the ground-truth signal.

Figure 5b shows input signals of sub-discriminators and the magnitude of their frequency responses. In the case of MPD, the frequency responses of input signals are not distorted except for aliasing. On the other hand, the input signals of MSD are getting smoother whenever down-sampling. It is due to the low-pass filtering of average pooling, which results into diminishing amplitudes in high frequencies.

When comparing the outputs of learned generators, the difference is more evident. Figure 5c shows the outputs of generators each of which is trained with MSD and MPD, respectively. In the case of MPD, the synthesized signal is similar to the ground truth, while in the case of MSD, the synthesized signal is noisy.

The reason of the sub-optimal outcomes of MSD can be found in average pooling of the down-sampling procedure. Figure 5d shows the down-sampled synthesized signals for sub-discriminators and the magnitude of their frequency responses. When it comes to down-sampling synthesized signals for the second and third sub-discriminators of MSD, the result seems similar to that of the ground-truth signal. It indicates that even though there is huge difference between the ground-truth and generated signals, the average-pooled input signals of some sub-discriminators of MSD can be similar. On the other hand, MPD shows similar results to the ground truth regardless of down-sampling factors. In conclusion, we can see that MPD captures more periodic patterns of an input signal than MSD, and capturing periodic patterns is important to model signals.

## Appendix C

### C.1. Details of architectural difference between RWD and MPD

When the first layer of RWD is the grouped 1d convolution, the convolution layer of RWD becomes similar to the first 2d convolution with k×1 kernels of MPD, but there is still a difference as RWD would not share weights across each group. Moreover, RWD operates on an entire input sequence regardless of input reshaping, while MPD only operates on down-sampled, or evenly spaced, samples from the input sequence. In that regard, it can be seen that RWD uses additional parameters to mix representations of down-sampled samples at each layer, in other words, MPD shares more parameters at each layer than RWD.