[Reviews · NeurIPS 2020]

Review 1

Summary and Contributions: This work proposes a GAN approach to synthesizing high quality speech waveforms. They also show that the audio is generated quickly on modern GPUs.

Strengths: This paper describes a few changes to current state-of-the-art GANs for audio synthesis that are claimed to improve quality. Specifically, the differences from MelGAN and GAN-TTS are (a) the existence of a mel-spectrogram loss and (b) the "MPD" multi-period discriminators. They do ablation studies on these and compare MOS to MelGAN, WaveGlow, and WaveNet.

Weaknesses: It's unclear why the changes help and whether they really do help. In particular, they claim that "When the model was trained without the mel-spectrogram loss, the training process became unstable, and some pronunciations were synthesized differently from the ground truth audio.". However, this is not the case for MelGAN or GAN-TTS, where a spectrogram L1/L2 loss is not necessary, which leads me to believe that the architecture used may not be well tuned. Additionally, it's unclear why MPD discriminators are needed instead of MSD discriminators; in fact, MPD discriminators seem strictly less powerful. Again, both GAN-TTS and MelGAN function well with only MSD discriminators, so it does not make sense that having only MSD discriminators would break the system.

Correctness: Overall, although the methodology is sound, it is unclear whether the results here are better or comparable to previous results. In particular, it seems like the claim being made is that GAN-TTS and MelGAN can be improved with some changes -- but in this paper, the changes seem *necessary* for them to work, which is empirically not the case.

Clarity: The paper is well written.

Relation to Prior Work: It is clear how this paper differs from prior work.

Reproducibility: Yes

Additional Feedback: It would be good to explain that MOS numbers for WaveNet, WaveGlow, and MelGAN are from your own models, *not* sourced from the original papers. It's unclear if that's the case, and if the scores are from the respective papers rather than your own models, it means the numbers may not be comparable. The biggest issue I have with this paper is the discrepancy between what this paper claims (STFT loss is critical to performance, MPD discrimiantors are important and MSD is not good enough) and what prior art and experience shows. Since the key contributions of this paper are specifically those changes, I have concerns about this paper. However, if I've misinterpreted the claims or results of the paper, I would like to understand how I have done so.


Review 2

Summary and Contributions: The paper proposes some improvements to MelGAN [1] (an adversarial model for mel-spectrogram inversion), mostly based on the discriminator architecture of GAN-TTS [2] and incorporating L1 spectrogram loss. These lead to higher fidelity scores in mel-spectrogram inversion and in speech synthesis when coupled with a pre-trained Tacotron2 and fine-tuned. In addition, the HiFi-GAN allows faster synthesis, and its the medium-scaled version (<1M parameters) outperforms the main baseline MelGAN model. ************************************** Update: I would like to thank authors for the rebuttal. After reading other reviews and authors' answers I decided to keep my score unchanged. It is still unclear for me what is the main novelty in this work. Although the results are promising, I consider the MPD architecture to be an iteration on the RWD architecture, and from the provided ablations I cannot tell if the improvements come from the prime reshape factors, kx1 2D-convolutional filters, or simply more parameters. In addition, lack of comparison against RWDs and ablations of the weakest V3 variant only make the judgement even more difficult. I suggest authors improve the experimental part and fix these deficiencies.

Strengths: The paper provides empirical evaluation of the proposed architecture and clear ablation study. Most components of the model are not novel: in particular, Multi-Period Discriminator (MPD) and Multi-Scale Discriminator (MSD) are special cases of Random-Window Discriminators (RWDs) introduced in GAN-TTS [2]. Apart from the L1 spectrogram reconstruction loss, the components of the training objective were explored in MelGAN [1]. On the other hand, the Multi Receptive Field Fusion module is a novel generator architecture which allows wide receptive fields.

Weaknesses: To achieve high fidelity speech waveforms, HiFi GAN requires either ground truth audio or fine-tuning. Scores for the actual speech synthesis for unseen text (with Tacotron2 used for generation spectrograms) look meagre. It also requires considerably longer training than comparable models (e.g. [2])

Correctness: Some of the novelty claims are incorrect (see 'Relation to prior work'). In addition, MOS test were carried out on a very small sample of 50 utterances, much smaller the commonly used 1000 utterances. The MOS scores are presented in a somewhat misleading way. It seems that the Table 1. presents results obtained via inversion of the ground truth spectrograms, which have limited relevance for the community and make comparisons with other works (such as WaveRNN [3], ParallelWaveNet [4], GAN-TTS [2]) even more difficult (of course works that use different datasets and MOS evaluation settings are not exactly comparable, yet this should not be exacerbated by methodology issues). I believe that Table 4. should be presented as the main result, as it shows the actual fidelity of the utterances synthesised from unseen text.

Clarity: The paper is easy to read in general; however, there are some unclear parts, for instance: lines 62-63: "a module (...) that places multiple residual blocks, each of which observes patterns of various lengths in parallel" line 65: "at 3.7 MHz" I guess it is related to synthesis speed, and not the frequency of synthesised audio (which later is specified as 22kHz). However it is very misleading. line 74: "two discriminators" it is inconsistent with Section 4.3 lines 126-127: "one scalar for each overlapping window" not sure what overlapping refers to. table 1.: It should be clarified that the MOS scores are for the inversion of ground-truth spectrograms. table 1 caption: "the model can generate n x 1000 samples per second" - if "samples" refers to timestemps, it can be quite misleading. table 2: Baseline (HiFi-GAN V3) has different score than in Table 1.

Relation to Prior Work: MPDs described in section 2.3 are described as completely novel architecture, yet they are closely related to the RWDs of GAN-TTS [2]. Authors frame their structure as "reshape to (T/p) x p" & "2d convolutions with 1xk kernels", which in fact is equivalent to RWD with grouped 1d convolutions. This correspondence has not been noted. Moreover, authors motivate MPDs in lines 96-99, claiming that periodic patterns in audio have not been addressed in the literature, which is not the case given existence and motivation for RWDs. The submission lacks comparisons to WaveRNN [3] and Parallel WaveNet [4], which are very relevant in the context of on-device synthesis, synthesis speed and numbers of parameters/FLOPs.

Reproducibility: Yes

Additional Feedback: [1] Kundan Kumar, Rithesh Kumar, Thibault de Boissiere, Lucas Gestin, Wei Zhen Teoh, Jose Sotelo, Alexandre de Brebisson, Yoshua Bengio, Aaron Courville, "MelGAN: Generative Adversarial Networks for Conditional Waveform Synthesis", NeurIPS 2019 [2] Mikołaj Bińkowski, Jeff Donahue, Sander Dieleman, Aidan Clark, Erich Elsen, Norman Casagrande, Luis C. Cobo, Karen Simonyan, "High Fidelity Speech Synthesis with Adversarial Networks", ICLR 2020 [3] Nal Kalchbrenner, Erich Elsen, Karen Simonyan, Seb Noury, Norman Casagrande, Edward Lockhart, Florian Stimberg, Aaron van den Oord, Sander Dieleman, Koray Kavukcuoglu, "Efficient Neural Audio Synthesis", ICML 2018 [4] Aaron van den Oord, Yazhe Li, Igor Babuschkin, Karen Simonyan, Oriol Vinyals, Koray Kavukcuoglu, George van den Driessche, Edward Lockhart, Luis C. Cobo, Florian Stimberg, Norman Casagrande, Dominik Grewe, Seb Noury, Sander Dieleman, Erich Elsen, Nal Kalchbrenner, Heiga Zen, Alex Graves, Helen King, Tom Walters, Dan Belov, Demis Hassabis "Parallel WaveNet: Fast High-Fidelity Speech Synthesis"


Review 3

Summary and Contributions: The paper introduces a novel discriminator called Multi-Period Discriminator (MPD), which is claimed to have better inductive bias to discriminate between real and fake audio waveforms since they are composed of periodic patterns. The proposed method reuses most other components from previous work. “Multi-Period Discriminator” is essentially a group of discriminators, each operating at subsampled waveforms (real or generated), where subsampling factor is carefully chosen (e.g. 2, 3, 5, 7, 11) so that each sub-discriminator look at different patterns. The effectiveness of the discriminator is tested on mel-spectrogram inversion tasks i.e. generating raw waveforms given the corresponding mel-spectrogram. Ablation study clearly demonstrates the effectiveness of MPD for the task.

Strengths: The proposed MPD is novel and well-motivated. It is effective as shown with empirical results and ablation study. Speech synthesis community (along with others who model periodic signals) will find value in this work.

Weaknesses: Since the core contribution of the paper is MPD, it should be substantiated by experiments on other datasets (even, some toy datasets) where periodic patterns exist. This is lacking in the current form of the paper.

Correctness: The empirical methodology could have been better: if the core contribution is MPD (or some other loss terms), this should be tested independent of any other changes in architecture. For example, one would expect to see improvement when MPD is introduced in the settings of other related works (GAN models) that authors compare with.

Clarity: The paper is sufficiently clear. It should clarify that the scope of the proposed idea and experiments is limited to mel-spectrogram inversion, not TTS in general.

Relation to Prior Work: Yes

Reproducibility: Yes

Additional Feedback: Authors should conduct additional experiments on other datasets with periodic signals. This can even be some toy datasets to bolster their claims. They should show some failure modes of the proposed model in order to specify the exact contribution of the proposed MPD.


Review 4

Summary and Contributions: The paper addresses the mel-spectrogram inversion problem, i.e. to generate raw waveform output from mel-spectrogram input using the GAN based network architecture. The main contribution of the paper is the proposal of a new model named HiFi-GAN for both efficient and high-fidelity speech synthesis, in which a set of small sub-discriminators have been designed. The design of the sub-discriminator is motivated by the fact that the periodic patterns is crucial for the generation of realistic speech audio which affects the speech quality. The mean opinion score (MOS) of the generated speech is quite close to that of human speech, outperforming compared state-of-the-art models such as WaveNet, WaveGlow and MelGAN. Furthremore, the proposed HiFi-GAN model has also good generality for unseen speakers.

Strengths: (1) The paper proposes a new model named HiFi-GAN for efficient and high-fidelity raw waveform generation from mel-spectrogram. (2) The idea of the design of the discriminator is new. In addition to the existing Multi-Scale Discriminator (MSD), the discriminator also consists of a set of small sub-discriminators (called Multi-Period Discriminator, MPD). Each MPD handles a portion of periodic signals of input audio to capture the diverse periodic patterns underlying in the audio data. (3) The design and introduction of MPD ensures the novelty of the proposed method. (4) Extensive experiments have been conducted to verify the effectiveness of the proposed model for the task of spectrogram inversion. (5) The experimental results are quite promising, with the new HiFI-GAN, the MOS of the generated speech is very close to the MOS of the natural speech recording (human voice), outperforming state-of-the-art models such as WaveNet, WaveGlow and MelGAN. (6) Ablation studies have been conducted to verify the contribution of different components, such as MPD, MRC, and mel-spectrogram loss. (7) Experimental results also validate the generality of the proposed method to unseen speaker. (8) The proposed HiFi-GAN, which is a kind of Vocoder, is also evaluated under the pipeline of end-to-end speech synthesis with good performance in MOS score. (9) The inference (synthesis) speed of HiFI-GAN is also promising. (10) The writing and presentation of the paper is clear and easy to follow.

Weaknesses: It would be better if the paper could provide more discussions to the related work, such as MelGAN, or Parallel WaveGAN with multi-resolution STFT loss, etc.

Correctness: (1) The correctness of the proposed method are proved by extensive experiments in Section 4. (2) The source code of the model is provided. (3) Considering the periodicity characteristics of the speech waveform, the introduction of Multi-Period Discriminator (MPD) is reasonable.

Clarity: The writing and presentation of the paper is clear and easy to follow.

Relation to Prior Work: (1) Experiments have been conducted to compare the proposed model with state-of-the-art models, such as WaveNet, WaveGlow, MelGAN, etc. (2) It would be better if the paper could provide more discussions to the related work, such as MelGAN, or Parallel WaveGAN with multi-resolution STFT loss, etc.

Reproducibility: Yes

Additional Feedback: Update: After reading the authors' feedback, and considering my previous review comments, I think the paper can be accepted. My detailed comments have been provided in the above review points. Related work available at: https://arxiv.org/abs/2006.05694

[Author Response · NeurIPS 2020]

We thank all the reviewers for their valuable comments.

**R1 :** We would like to clarify that, 'When the model was trained without the mel-spectrogram loss, the training process became unstable, and some pronunciations were synthesized differently from the ground truth audio.' only corresponds the result of the ablation studies using the V3 generator, which has the smallest expressive power among the three generator variations. The problem we mentioned is also present in MelGAN, and when referring to our additional experiment results[Table 1], it can be seen that the V3 generator performs better than MelGAN without L1 loss.(The details of additional experiments are described in the following section.) We understand that this unexplained part can be misleading, and we will make up for it in the final version. Additionally, MelGAN and GAN-TTS are studies that do not use L1/L2 loss, but there are several studies such as Isola et el.[1], Parallel WaveGAN, and MB-MelGAN to use L1/L2 loss. We also think that applying the L1/L2 loss gives no disadvantage in one-to-one mapping as our work. It is because the loss helps for the generated samples to be close to the target ground truths. All MOS numbers for WaveNet, WaveGlow, and MelGAN are from the publicly available implementation of the models, not sourced from the original papers. We will clarify the details of the experiments in Section 3.

**R1 & R3 :** To verify the effect of MPD in the settings of other GAN models, we introduced MPD in MelGAN and conducted MOS evaluations. Specifically, we trained MelGAN up to 500k steps and compared it with all samples used in ablation study as well as samples of V1. The summarized MOS evaluation results are shown in [Table 1]. MelGAN trained with MPD outperforms the original one by a gap of 0.50 MOS, which shows statistically significant improvement. Furthermore, considering that V1 and V3 generators outperform MelGAN, we claim that our overall architecture is well tuned. We will update the result of the extended ablation studies to our final version and add related audio samples at the bottom of the demo page.

Table 1: Mean Opinion Scores. All models were trained up to 500k steps.

| Model | MOS | 95% CI |
|---|---|---|
| Ground Truth | 4.40 | ±0.07 |
| MelGAN | 3.15 | ±0.12 |
| MelGAN + MPD | 3.65 | ±0.09 |
| HiFi-GAN V1 | 4.30 | ±0.07 |
| HiFi-GAN V3 | 4.11 | ±0.07 |
| HiFi-GAN V3 w/o L1 | 3.43 | ±0.10 |

**R2 :** As for MPD and RWD, leaving the differences such as existence of Markovian window or strided convolutions aside, there is resemblance in the initial convolutional layers as Reviewer 2 pointed out. When the first layer of RWD is the grouped 1d convolution, the 2d convolution with 1xk kernels of MPD becomes similar to RWD, but there is still a difference as RWD would not share weights across each group. In that regard, we argue that our 2d convolution operations in MPD help to increase parameter efficiency. We will add the similarity and difference of MPD and RWD in the paper.

As vocoders are only trained on ground truth mel-spectrograms, fine-tuning with predicted samples from TTS models helps to improve the overall mel-spectrogram-based end-to-end speech synthesis quality. Although Reviewer 2 commented scores for actual speech synthesis for unseen text look meagre, we believe we showed the strength of our model in that our model scored higher than the comparison model even before fine-tuning, and after fine-tuning, the quality improved further.

For MOS tests, we set WaveGlow, MelGAN and MoL WaveNet as our comparison group, because they use 1) commonly used mel-spectrograms as input conditions and 2) they are publicly available in the open source community. Since the mentioned models such as WaveRNN, ParallelWaveNet, and GAN-TTS are reported to use different input conditions called linguistic features, which makes them hard to reproduce, we do not set them as our comparison group.

**R3 :** Thanks Reviewer 3 for the suggestions and concerns. We will conduct additional experiments to verify the effectiveness and limitation of MPD for capturing periodic patterns and add the results to the final version.

**R4 :** Following the valuable suggestions, we will provide more discussion about the related works, which Reviewer 4 mentioned, in the new version of the paper.

[1] Isola, Phillip, et al. "Image-to-image translation with conditional adversarial networks." Proceedings of the IEEE conference on computer vision and pattern recognition. 2017.

[Meta-Review · NeurIPS 2020]

This work initially received mixed reviews, but after the author feedback cleared up a misunderstanding, most reviewers are now recommending acceptance. Nevertheless, I think R2 (who has not raised their score) has some valid concerns, which I want to account for in my decision. I have decided to recommend acceptance. The experimental section of this work is fairly comprehensive, and adequately demonstrates that the proposed architecture is effective. However, it is important to point out that the majority of experiments was conducted using ground-truth mel-spectrogram conditioning, which does not match the usual practical setting of TTS systems, where the spectrograms are themselves generated by a model (and thus imperfect). I would encourage the authors to make this abundantly clear in the manuscript, and to consider including additional experiments in the "true" TTS setting to balance things out. R2 points out that both MPD and MSD architectures show a lot of similarities to the random window discriminators from the GAN-TTS paper. I think this is a fair observation, and it would be helpful to discuss the differences in more detail in the updated manuscript (e.g. the weight sharing after initial "reshape", using prime downsampling factors). The authors have committed to this in their feedback. In particular, it is important to accurately convey which ideas are novel and which have previously been proposed in literature, so the reader does not come away with the wrong impression. Furthermore, any claim that these architectural choices have a better inductive bias for periodic signals should be properly motivated, both theoretically and empirically. R2 also suggests an additional ablation comparing MPDs with prime and non-prime factors, leaving everything else unchanged, and I think this would be a useful addition. I concur with R1 that the choice of baselines for this work is appropriate, and the addition of an experimental comparison to WaveRNN or Parallel WaveNet is not a requirement to meet the bar for acceptance. That said, the differences with these approaches should at least be discussed qualitatively in the context of fast real-time synthesis (I believe this is already the case to some extent for Parallel WaveNet, but not WaveRNN). One small note regarding the author feedback: the original WaveNet model did not use mel-spectrogram conditioning -- vocoder variants of WaveNet were only introduced in later works. I wanted to point this out, in case the authors were intending to reuse this part of the author feedback in their manuscript. Given the recent publication of a paper with a very similar name (https://arxiv.org/abs/2006.05694), as pointed out by R4, the authors may also want to consider a name change. This is merely a suggestion, not a requirement on our part.